# *TaCKX2.2* Genes Coordinate Expression of Other *TaCKX* Family Members, Regulate Phytohormone Content and Yield-Related Traits of Wheat

**DOI:** 10.3390/ijms22084142

**Published:** 2021-04-16

**Authors:** Bartosz Jablonski, Karolina Szala, Mateusz Przyborowski, Andrzej Bajguz, Magdalena Chmur, Sebastian Gasparis, Waclaw Orczyk, Anna Nadolska-Orczyk

**Affiliations:** 1Department of Functional Genomics, Plant Breeding and Acclimatization Institute—National Research Institute, Radzikow, 05-870 Blonie, Poland; b.jablonski@ihar.edu.pl (B.J.); k.szala@ihar.edu.pl (K.S.); m.przyborowski@ihar.edu.pl (M.P.); s.gasparis@ihar.edu.pl (S.G.); 2Laboratory of Plant Biochemistry, Faculty of Biology, University of Bialystok, Ciolkowskiego 1J, 15-245 Bialystok, Poland; abajguz@uwb.edu.pl (A.B.); m.chmur@uwb.edu.pl (M.C.); 3Department of Genetic Engineering, Plant Breeding and Acclimatization Institute—National Research Institute, Radzikow, 05-870 Blonie, Poland; w.orczyk@ihar.edu.pl

**Keywords:** *TaCKX2.2* GFMs silencing, wheat, phytohormone homeostasis, cytokinins, yield-related traits, chlorophyll content

## Abstract

*TaCKX* gene family members (GFMs) play essential roles in the regulation of cytokinin during wheat development and significantly influence yield-related traits. However, detailed function of most of them is not known. To characterize the role of *TaCKX2.2* genes we silenced all homoeologous copies of both *TaCKX2.2.1* and *TaCKX2.2.2* by RNAi technology and observed the effect of silencing in 7 DAP spikes of T_1_ and T_2_ generations. The levels of gene silencing of these developmentally regulated genes were different in both generations, which variously determined particular phenotypes. High silencing of *TaCKX2.2.2* in T_2_ was accompanied by slight down-regulation of *TaCKX2.2.1* and strong up-regulation of *TaCKX5* and *TaCKX11*, and expression of *TaCKX1*, *TaCKX2.1*, and *TaCKX9* was comparable to the non-silenced control. Co-ordinated expression of *TaCKX2.2.2* with other *TaCKX* GFMs influenced phytohormonal homeostasis. Contents of isoprenoid, active cytokinins, their conjugates, and auxin in seven DAP spikes of silenced T_2_ plants increased from 1.27 to 2.51 times. However, benzyladenine (BA) and abscisic acid (ABA) contents were significantly reduced and GA_3_ was not detected. We documented a significant role of *TaCKX2.2.2* in the regulation of thousand grain weight (TGW), grain number, and chlorophyll content, and demonstrated the formation of a homeostatic feedback loop between the transcription of tested genes and phytohormones. We also discuss the mechanism of regulation of yield-related traits.

## 1. Introduction

Bread wheat (*Triticum aestivum* L.) is one of the most widely grown and economically important cereal species in the world. This high-yielding species is comparable in production to rice and maize but contains higher protein level, and is rich in vitamins and dietary fibres [1,2]. A further increase of wheat yield is important to feed the growing world population [3,4]. The large and complex, allohexaploid genome of bread wheat, which is composed of three diploid genomes (AABBDD), is a great reservoir of homoeologous genes. Knowledge of their function can be applied directly or after genetic modifications in wheat breeding as reviewed previously [5]. Therefore, functional characterization of yield-related genes in this species is especially important. Transformation technology used in this type of research is still a bottle-neck in the case of species with a large and complex genome. This is a possible reason for the weak progress in research on bread wheat compared to other cereals. 

Cytokinin (CK) is an important growth regulator of plant development influencing many yield-related processes as reviewed in Kieber and Schaller [6] and in Sakakibara [7]. This phytohormone is considered as ‘a key driver’ of seed yield in many crops [8] including barley [9,10] and wheat [11,12,13]. Its function is coordinated by various phytohormones: auxin, gibberellins (GAs), brassinosteroids and others [14]. Generally, cytokinin promotes cell division, and auxin cell expansion and the balance between these two processes controls root and shoot growth. Biosynthesis of gibberellins is supported by auxin transport. Therefore, GAs repress tillering but promote stem elongation. Tillering is inhibited by auxin, as well as by reducing CK content [12,15], reviewed in Guo et al. [14]. Moreover, CKs are negative regulators of resistance to abiotic stress and root growth [16,17] and reduction of their level increases ABA accumulation [18]. However, in wheat kernels an increased *cis*-zeatin (cZ) level was followed by increased ABA accumulation [13]. Hormone homeostasis is further regulated by co-expressed genes and other factors including miRNA and signalling proteins. Integrators of these various developmental signals are transcription factors, which might reprogram expression of the interacting genes [19]. Fine tuning the expression of key genes’ might be a promising strategy to improve yield-related traits [14].

Cytokinin levels in tissues and organs of plants is regulated by biosynthesis and metabolism as well as compartmentalization and translocation through the plant [8,20]. Cytokinin metabolites differ in their biological activity [21,22]. Among different forms of CKs, the most abundant are isoprenoid CKs: N^6^-(Δ^2^-isopentenyl)adenine (iP), *trans*-zeatin (tZ), *cis*-zeatin (cZ) and dihydrozeatin (DZ). Benzyladenine (BA) belongs to the second smaller group of aromatic CKs [23]. These free base CKs listed above, which are products of biosynthesis, are the most biologically active. They undergo a series of interconversions to nucleosides (CK ribosides) and nucleotides as summarized in Hosek et al. [24]. Nucleosides-tZ riboside (tZR), cZR, DZR and iPR-are believed to be the main transporter metabolites, although CK bases, tZ and iP were found in xylem sap as well [20]. Translocation of tZ and tZR via the xylem from roots to shoots controls various shoot traits [7]. Conjugation of free bases with glucose via *O*-glucosyltransferases leads to their rapid deactivation to *O*-glucosides, which can be converted back to free bases and ribosides by β-glucosidase. This enzyme was identified in maize [25], products were detected in 7 days after pollination (DAP) spikes of wheat [11] and alleles of *TaZOGs* and *cisZOGs* were localized [26] and updated (J. Song, personal communication). Another enzyme, N-glucosyltransferase glycosylate bases of all CK-types at the N7 and N9 positions, making them biologically inactive in Arabidopsis [27]. However, N9-glucosides of iP, tZ and cZ were reported to be substrates of cytokinin oxidase/dehydrogenase enzyme in maize [28] and tZ-N7G, tZ-N9G in Arabidopsis [24]. 

Cytokinin oxidase/dehydrogenase (CKX) catalyzes irreversible degradation of free bases iP, tZ, cZ and their ribosides iPR, tZR, cZR by cleaving the *N*^6^-side chain. DZs are resistant to this enzyme [23]. This step of CK metabolism plays an important role in the regulation of cytokinin level and regulation of yield-related traits in cereals, inter alia rice, barley and wheat, as was documented by RNAi silencing of selected genes [9,10,11,29,30]. 

CKX enzymes are encoded by *CKX* gene family members (GFMs). According to the latest revision by Chen et al. [12] there are 13 *TaCKX* GFMs basically numbered from one to 11, and 12 of them are located on the three homoeologous chromosomes of the A, B and D genome, giving the number of 35 GFMs. Exception are *TaCKX2.2.2* and *TaCKX2.2.3* allocated only to the D genome. This new numbering is based on the newly released wheat genome database (IWGSC RefSeq v1.0 & v2.0) and phylogenic cladogram with other cereals (barley and rice). The numbering of *TaCKX2*, which has undergone gene duplication [31], is the most expanded. The previous number of *TaCKX2.2* [32] was revised for *TaCKX2.2.1* located on A, B, and D genomes and mentioned above *TaCKX2.2.2* and *2.2.3*. Other genes revised according to previous publications [26,32], which are referred to this study, are *TaCKX10* renamed as *TaCKX9* and *TaCKX3* renamed as *TaCKX11*. The *TaCKX* GFMs are specifically expressed in various tissues and organs [12,32,33], which might indicate their role in reproductive development [10]. According to the last assignments, *TaCKX1*, *TaCKX2.2.1* and *TaCKX2.2.2* are mainly expressed in developing spikes, *TaCKX3* and *TaCKX10* in seedling roots, *TaCKX5*, *TaCKX9* and *TaCKX8* in younger organs of developing plants from seedling roots to 0 days after pollination (DAP) spikes, and *TaCKX11* and *TaCKX4* in all organs [32,33]. Similar tissue specificity of expression of *TaCKX* GFMs was presented based on RNA-seq data [12], as well as for *TaCKX1* and *TaCKX2* during early kernel development [13].

Only some *TaCKX* GFMs are functionally characterized. Silencing of *TaCKX1* by RNAi influenced levels of expression of the other *TaCKX* GFMs, and increased content of most CK forms and GA_3_, which caused an increase of yield parameters such as a higher spike number, grain number, and grain yield, but lower thousand grain weight [11]. The effect of *TaCKX2.4* gene silencing by RNAi increased grain number per spike [34]. Another source of analysis of gene function and/or selection of desirable phenotypes in bread wheat, for which natural or induced mutants are not available, is natural variation among genotypes [33]. Haplotype variants of *TaCKX6a02* and *TaCKX6-D1* were associated with higher filling rate and grain size [35,36], and higher copy number of *TaCKX4* was related to increased chlorophyll content and grain size [37]. The above-mentioned genes are now annotated: *TaCKX6a02* as *TaCKX2.2.1–3A*, *TaCKX6-D1* (JQ797673) as *TaCKX2.2.2–3D* and *TaCKX2.4* as *TaCKX2.2.1–3B* [12].

Some functions of these genes might also be predicted by known function of their orthologues in other cereal species. Silencing of *HvCKX1* orthologous to *TaCKX1* resulted in elevated grain number in these two, closely related species [9,11]. However, the same effect of increased grain number in rice was obtained by silencing or knocking-out *OsCKX2* [30]. *OsCKX11*, orthologous to wheat *TaCKX11*, which is expressed in all organs of the wheat plant [12,32], coordinated sink–source relationships and leaf senescence and grain number in rice [38]. 

Both methods of gene silencing by RNAi and knock-out mutation by CRISPR/Cas9 have been successfully used for analysis of gene function. The advantages of the first one are the possibility of silencing homoeologous copies of the genes and obtaining plants with different levels of silencing. This is especially important in the case of developmentally regulated genes, for which the level of silencing might lead to different phenotypes [11], or in the case of genes coding proteins of key importance for life [39]. The phenotypic differences between silencing of *HvCKX1* [9] and knocking-out the same gene [40] are documented.

In this study we silenced homoeologues of both *TaCKX2.2* genes, *TaCKX2.2.1* and *TaCKX2.2.2*, by RNAi technology and analyzed the phenotypic effect of silencing them in T_1_. In addition to these two gene family members we monitored *TaCKX2.1* and other *TaCKX* GFMs in the T_2_ generation. Similar study has already been performed in our lab with another GFM, *TaCKX1*, silenced in the same genotype [11] and the data are compared. To reveal mechanisms of regulation of yield-related traits cross-talk of silenced genes with other *TaCKX* GFMs as well as homeostatic feedback loop between their transcription and phytohormones was investigated. 

## 2. Results

### 2.1. Expression Levels of Silenced TaCKX2.2.1, TaCKX2.2.2 and TaCKX2.1 in T_1_ and T_2_

The silencing cassette contains homologous sequences for silencing of three *TaCKX2* genes, *Ta**CKX2.2.1*, *Ta**CKX2.2.2* and *TaCKX2.1*. Therefore, the silencing effect for the two was tested in T_1_ (Figure 1a,c) and a test for *Ta**CKX2.1* was included in the T_2_ generation (Figure 1d,e). The range of relative expression levels for *Ta**CKX2.2.1* in T_1_ was from 0.42 to 1.08, and the mean was 0.89. The mean for silenced plants (≤0.8) was 0.65. Relative expression levels for *Ta**CKX2.2.2* ranged from 0.41 to 1.66 and the mean for silenced plants was 0.68. Silencing for both genes in tested T_1_ plants was similar for those with higher silencing but differed in the plants with higher expression levels. The correlation coefficient between the levels of expression of *Ta**CKX2.2.1* and *Ta**CKX2.2.2* was 0.53 for 77 plants (N 77).

The most silenced gene in T_2_ was *Ta**CKX2.2.2* (Figure 1d,e). Levels of silencing of this gene in tested lines ranged from 0.26 to 1.25 and the mean for the highly silenced one (≤0.60) was 0.49; for those with relative expression between 0.61 to 0.79 it was 0.7, and for the third group of non-silenced genes it ranged from 0.8 to 1.25 with a mean of 0.96 (see also Figure 2 and Appendix A). Levels of expression of *Ta**CKX2.2.1* ranged from 0.44 to 2.01 with the mean of 1.14, but silencing below 0.79 was observed only in six out of 62 tested plants. Expression levels of *Ta**CKX2.1* ranged from 0.69 to 2.01, the mean was 1.17 and only in three plants the expression level was below 0.79.

### 2.2. Yield-Related Traits are Determined Depending on the Levels of Silencing of TaCKX2.2.1 and TaCKX2.2.2

The T_1_ silenced plants were divided into two groups. The first one included *Ta**CKX2.2.2* silenced plants (level of expression 0.41 to 0.77; mean 0.68; Figure 2a, Appendix Aa), and the second contained *Ta**CKX2.2.1* silenced plants (level of expression 0.42 to 0.79; mean 0.65; Figure 2b, Appendix Ab). In both groups, expression of the second, respective *Ta**CKX2.2* gene was silent at the level of 0.59 or slightly silenced (0.90) respectively. Data of yield-related traits in both groups of silenced plants were compared to those in non-silenced groups of plants with the expression level from 0.8 to above 1.00. In *Ta**CKX2.2.2* silenced plants, spikes were shorter and data of plant height, spike number, spike length, grain number, grain yield, thousand grain weight (TGW), and chlorophyll content measured by SPAD in flag leaves of the 1st spike (SPAD 1st) were slightly lower compared to the non-transgenic control (Figure 2a, Appendix Aa). Otherwise CKX activity was slightly higher (Appendix Aa). Different data were obtained in *Ta**CKX2.2.1* silenced plants (Figure 2b, Appendix Ab). These plants were significantly taller and the values of spike length, grain number, grain yield and TGW were slightly higher compared to the non-transgenic control.

Among 65 T_2_ analyzed plants (Figure 2c, Appendix Ac) the level of *TaCKX2.2.2* expression in 25 of them was very low (0.26–0.60), in 23 of them was moderate (0.61–0.79) and in 17 of them it was from 0.80 to 1.25 (non-silenced plants). Additionally, 18 plants were not-silenced, R_2_ in-vitro control (second generation of in-vitro regenerated plants). Relative expression levels of *Ta**CKX2.2.1* and *Ta**CKX2.1* in these three groups of T_2_ plants were above 1.00 and relative CKX activity was slightly lower in the most silenced (0.94) and higher in non-silenced plants. Similarly, as in the case of T_1_ plants with silenced *Ta**CKX2.2.2*, the data of plant height, spike number, spike length and grain number were lower compared to the control group (Figure 2c, Appendix Ac). However, the data of TGW and SPAD in the first and in the next spikes were significantly higher, especially in the first, most silenced group compared to the control.

### 2.3. Co-Expression of Silenced TaCKX2.2.2 with Other TaCKX GFMs in T_2_

Co-expression of other genes was measured in 7 DAP spikes of seven T_2_ plants representing the highly silenced *TaCKX2.2.2* (Figure 3). The mean level of silencing for *TaCKX2.2.2* compared to the control (=1.00) was 0.42. Expression levels of *TaCKX1*, *TaCKX2.1* and *TaCKX9* were close to 1.00 (0.87, 0.88 and 1.05 respectively). For *TaCKX2.2.1*, the level was 0.80, and for two of them, *TaCKX5* and *TaCKX11*, levels were much higher, reaching 2.34 and 1.64 respectively. CKX activity was slightly lower.

### 2.4. Phytohormone Content in Highly Silenced TaCKX2.2.2 T_2_ Lines

Phytohormone content was measured in 7 DAP spikes of seven T_2_ plants representing the highly silenced *TaCKX2.2.2* gene with mean relative expression 0.42 (Figure 4a,b). The highest content in both silenced and non-silenced controls was observed for tZ and tZGs, cZ and cZOG, DZGs and ABA. These contents were even higher for almost all listed cytokinins in silenced plants, apart from BA (and ABA). A larger increase in content was noted for tZGs, cZ, and DZR. The latter was not present in the non-silenced control. Interestingly, BA and ABA contents in silenced plants were decreased and GA, detected only in non-silenced, control plants was reduced to 0. 

Small amounts (≤0.20 ng/g biomass) were detected for DZOGR, DZ7G, cZOGR, IP7G; trace amounts (≤0.02 ng/g biomass) for: cZ9G, CZR DZ9G; and DZ, iPR, IBA, IPA, and PAA were not detected.

The phytohormone ratio indicator presented in Figure 4b is the mean value in silenced per mean value in non-silenced, control plants indicating a decrease or increase in content above the control level.

### 2.5. Co-Ordinated Regulation of CKX2.2.2 Silencing with Expression of Other TaCKX GFMs, Phytohormone Content and with Yield-Related Traits

A schematic presentation of co-ordinated regulation of expression of *TaCKX* GFMs, phytohormone content and yield-related traits is presented in Figure 5. Significantly decreased expression of *TaCKX2.2.2* (58% less than in control) in 7 DAP spikes of silenced T_2_ plants was co-ordinated with slightly decreased expression of *TaCKX2.2.1* (20%) and highly increased expression of *TaCKX5* and *TaCKX11* (234% and 164% respectively). Relative expression levels of *TaCKX1*, *TaCKX2.1*, and *TaCKX9* were close to the control (±5−13%). Most of isoprenoid cytokinin levels measured in seven DAP spikes of the same plants were increased. There were tZ (127%), tZGs (173%), tZR (203%), cZ (214%), DZGs (162%), DZR (251%), and iP (1.73%). cZOG content was not changed (103%). Content of aromatic cytokinin, BA was significantly reduced to 17% of control plants. The contents of other phytohormones were also different in silenced compared to control plants; IAA content was increased to 232% of the control, GA was not detected and ABA content reduced to 52% of control plants. All these changes significantly influenced three out of eight yield-related traits. TGW in silenced plants was significantly increased to 147%, grain number was reduced by 24% and flag leaf senescence was increased to 126%. Plant height, spike number and length, grain yield, and root weight were not changed compared to the control plants.

Correlation coefficients between *TaCKX* GFMs and phytohormones are presented in Table 1. Down-regulated *TaCKX1* and *TaCKX2.2.1*, unchanged *TaCKX9* and up-regulated *TaCKX5* expression are negatively correlated with tZR and IAA. Decreased expression of *TaCKX2.1* is strongly and negatively correlated with DZR as well as with tZGs and iP. Down-regulated *TaCKX2.2.2* is strongly, positively correlated with ABA, but up-regulated *TaCKX11* is negatively correlated with ABA.

### 2.6. Effect of TaCKX GFMs on Regulation of Phytohormone Content and Yield-Related Traits

The effect of *TaCKX* GFMs expression levels on regulation of phytohormone content and yield-related traits is based on correlation coefficients (cc; all collected in Appendix A) and presented in Figure 6.

Plant height is in positive correlation with grain yield, TGW, spike length and grain number positively correlated with *TaCKX9* in seven DAP spikes of T_2_ plants silenced for *TaCKX2.2.2*, and negatively with growing content of tZR and unchanged cZOG. Expression level of *TaCKX9* negatively correlated with tZR and cZOG as well. Moreover, plant height was also correlated with CKX activity, which is negatively correlated with cZ content.

An increase in silenced plants’ TGW is positively correlated with plant height as well as SPAD2. The trait is strongly, negatively correlated with slightly decreased expression of *TaCKX2.1* and positively with increased content of iP, tZGs and DZR. However, expression level of *TaCKX2.1* is negatively correlated with these phytohormones. In contrast, TGW is positively correlated with *TaCKX9* and CKX activity and negatively with tZR and cZ. The gene was positively correlated with tZR and CKX activity negatively correlated with cZ.

Grain number reduced in silenced plants is positively correlated with grain yield, plant height, spike number, and spike length (cc 0.90, 0.85, 0.77, and 0.68 respectively). The trait is positively correlated with strongly decreased expression of *TaCKX2.2.2* and negatively with cZOG but positively with decreased contents of BA and ABA. *TaCKX2.2.2* is also negatively correlated with cZOG but positively with ABA (cc 0.83) and BA (cc 0.58).

Grain yield is not changed in silenced compared to non-silenced plants. The trait is positively correlated with grain number (0.90), plant height (0.85), spike number (0.77), and spike length (0.68), and with *TaCKX9*, but negatively with tZR and cZOG. Correlation coefficients between *TaCKX9* expression and both cytokinins were negative (−0.83 and −0.59). Also grain yield is positively correlated with reduced contents of BA and ABA.

Spike length is in positive correlation with grain number and yield, plant height and spike number. The trait is positively correlated with *TaCKX2.2.2* and strongly, negatively with cZOG (cc-0.85) but strongly, positively with ABA (cc 0.94). Interestingly the gene is also negatively correlated with cZOG but strongly, positively correlated with ABA (cc 0.83).

Spike number, positively correlated with other traits such as grain number and yield, spike length and negatively with chlorophyll content measured in flag leaves (SPAD2). The trait was not correlated with expression of any gene, but was positively correlated with BA (cc 0.83), ABA (cc 0.64), and cZ (cc 0.62). 

Seedling root weight, which is negatively correlated with SPAD2, was strongly, negatively regulated by *TaCKX2.2.1* (cc −0.83) and positively with tZR and tZGs measured in seven DAP spikes. The gene is negatively correlated with both tZ metabolites. Root weight is also positively regulated by *TaCKX2.2.2*.

SPAD2 negatively correlated with spike number, grain number and root weight but positively with TGW. The trait is strongly, negatively correlated with *TaCKX2.1* and *TaCKX2.2.2* as well as with BA and ABA. Moreover, both genes positively correlated with the contents of both phytohormones.

## 3. Discussion

### 3.1. Phenotype Is Dependent on the Level of TaCKX2.2.2 and TaCKX2.2.1 Silencing

Silencing of particular genes essentially influenced the phenotype of wheat plants. Decreased expression of both, *TaCKX2.2.2* and *TaCKX2.2.1* by 41% in the T_1_ group of plants resulted in obtaining shorter plants and slightly lower parameters of yield-related traits. The opposite phenotype was obtained for plants, in which *TaCKX2.2.1* expression was decreased and *TaCKX2.2.2* expression was comparable to the control. These results indicate different functions of the two genes. Different phenotypes were also obtained with slightly silent (reduced by 30%) and highly silent (reduced by 51%) expression of *TaCKX2.2.2* in T_2_. A similar dependence of yield-related traits on the level of silencing of another GFM, *TaCKX1*, has already been documented in our earlier work with the same genotype [11]. Different levels of silencing of particular gene resulted in up or down-regulation of expression of other *TaCKX* genes as well as appropriate phytohormone homeostasis (discussed below). 

### 3.2. TaCKX2.2.2 Silencing Co-Ordinates Expression Levels of Other TaCKX GFMs and Phytohormone Homeostasis, Influencing Yield

A high level of *TaCKX2.2.2* silencing (in T_2_) was coordinated with a slight decrease of *TaCKX2.2.1* expression but a large increase of *TaCKX5* and *TaCKX11* expression. Otherwise, the effect of silencing of *TaCKX1* coordinated expression of the same genes in the opposite way [11]. Silencing of both *TaCKX2.2* GFMs, which are specifically expressed in developing spikes, caused slight silencing of *TaCKX2.1* and *TaCKX1* expressed in the same organ, as well. Increased expression was observed for *TaCKX5*, which is expressed in younger organs (from seedling roots to 0 DAP spikes) as well as *TaCKX11* expressed in all organs up to 14 DAP [32]. Growth of expression of these genes in 7 DAP spikes of plants representing decreased expression of *TaCKX2.2* genes specific for developing spikes might be a mechanism of maintaining CKX isozyme homeostasis, since total CKX activity was not changed. A similar result was observed in *TaCKX1* silenced plants [11]. In these plants down-regulation of expression of *TaCKX1* was coordinated with down-regulation of expression of *TaCKX11* and up-regulation of *TaCKX2.1* and *TaCKX9*, but total CKX activity was as in non-silenced plants. 

As expected, in both cases of *TaCKX2.2.2* and *TaCKX1* silencing, most cytokinins and their conjugates in seven DAP spikes increased. However, there are significant differences in the contents of various forms. The main differences include highly increased content of tZ and DZ ribosides in *TaCKX2.2.2* silenced compared to *TaCKX1* silenced plants, in which contents of these ribosides were very small or not detected [11]. Other differences concerning CKs are a larger increase of cZ and glucosides of tZ and DZ in *TaCKX2.2.2* silenced compared to *TaCKX1* silenced plants, which in contrast were very rich in tZ7G and tZ9GOG. Both riboside forms and free base cZ are transporter metabolites, which when transported from roots to shoots control shoot traits [7,20]. Moreover, tZ riboside and cZ are degradation substrates of CKX [23,24]. Therefore, their growth might be the joint effect of transport as well as decreased CKX activity of specific isozyme encoded by silent *CKX2.2.2*. Glycosylation conjugates are deactivation products of free bases CKs, and might be converted back to free bases and ribosides during reactivation [24,25]. Their growth of content in seven DAP spikes of *TaCKX2.2.2* silenced plants is possibly a result of maintaining/regulation of cytokinin homeostasis, to decrease intensive growth of free base tZ, cZ and their ribosides. Some of the *TaGlu* genes encoding β-glucosidases, the enzymes catalyzing the process of reactivation of *O*-glucosides to free bases and ribosides, were highly expressed in developing spikes of wheat, with the highest expression of *TaGlu4* in 7 DAP spikes [26].

Other differences in the effect of silencing of *TaCKX2.2.2* versus *TaCKX1* were associated with changes in IAA and GA_3_ contents as well as phenotype. IAA in silenced *TaCKX2.2.2* was 1.45 times higher, but GA content was reduced to zero. Conversely, IAA in silenced *TaCKX1* plants was not changed and GA was over 10 times higher [11]. The opposite effect of coordinated expression in the case of both *TaCKX2.2.2* and *TaCKX1* as well as CKs, auxin, ABA and GA resulted in opposite phenotype changes. The *CKX2.2.2* silenced plants were characterized by higher TGW, lower grain number, and higher chlorophyll content in flag leaves. The *TaCKX1* silencing resulted in a higher spike number, grain number, and grain yield, but lower TGW [11]. The possible mechanisms of regulation of these phenotypic traits are discussed below.

### 3.3. Homeostatic Feedback Loop Between TaCKX Genes Transcription and Phytohormones

Expression of Arabidopsis *AtCKX4* [41] activated by cytokinins was found to be down-regulated by IAA [42], underlining its importance in auxin-cytokinin crosstalk [43]. A similar feedback loop was observed in *TaCKX2.2.2* silent wheat plants. Down-regulation of this gene together with *TaCKX2.2.1* and *TaCKX1* in developing spikes influenced growing content of active forms of CKs, mainly cZ and iP as well as ribosides of tZ and DZ. Such a high *CKX* gene specific affinity to cZ was also shown in Arabidopsis by *AtCKX1* and *AtCKX7* [44]. Moreover, iP and tZ were found as the main substrates of *AtCKX4* [44], but it was associated with abiotic stress tolerance [42]. Activated by these cytokinins, highly expressed *TaCKX5* showed a strong, negative correlation with IAA. As in the case of *AtCKX4*, *TaCKX5* was highly expressed in younger organs (from seedling roots to 0 DAP spikes) but both *TaCKX2.2* genes are expressed in developing spikes [32]. Therefore, these three *TaCKX* genes take part in maintaining a homeostatic loop between CKs and IAA in developing wheat plants. As reviewed by Cao et al. [45], auxin was proven to be a key regulator of different processes involved in seed development, seed size and weight. Seed development has been influenced by auxin level, their transport and signalling pathway and mediated by auxin homeostasis. IAA was also found to stimulate transport of photoassymilates to and within developing grains of wheat [46]. 

ABA level was positively correlated with *TaCKX2.2.2* expression and negatively with cZOG. Therefore, the down-regulation of this gene, which is specifically expressed in developing spikes, was associated with decreased ABA and unchanged cZOG level. Moreover, high level of DZ riboside negatively correlated with slightly down-regulated *TaCKX2.1* but positively with high level of tZ glucosides. High tZ riboside level was negatively correlated with *TaCKX9* expressed through the younger organs of developing wheat plants.

Homeostatic balance of phytohormones in a selected organ is also regulated by many other processes. According to earlier reports, iP-type cytokinins might be converted to tZ-type cytokinins by cytochrome P450 mono-oxygenase proteins [43,47]. Both tZ and iP are also main, long-range transport forms. The tZ and its riboside are transported from root to shoot in the xylem, and iP is transported in the opposite direction through the phloem [48,49,50]. Possible consequences of the crosstalk between *TaCKX* genes and phytohormones in determination of yield-related traits are discussed below.

### 3.4. Regulation of Developmental Processes and Yield-Related Traits

Growing active CK content in 7 DAP spikes of *TaCKX2.2.2* silenced plants, which are in the middle of the cell division/cell expansion stage [51,52], means that both processes of cell differentiation and cell division in the embryo promoted by CKs [53] are active. In contrast, content of one kind of gibberellin measured, GA_3_ was reduced to 0. GA and ABA are considered to be the main regulators of seed germination and dormancy, playing antagonistic roles during this process. GA enhances germination while ABA positively regulates the induction and maintenance of dormancy [54,55]. The induction of seed dormancy by ABA takes place in the third phase of wheat seed development [56]. The seven DAP spikes of *TaCKX2.2.2* silenced plants showed decreased ABA content and GA_3_ content was not detected compared to non-silenced, control plants. These data might indicate a delay in seed germination and dormancy of silenced plants.

Silencing of *TaCKX2.2.2* influenced three out of eight yield-related traits testedgrowth of TGW, reduction of grain number per plant and increase of chlorophyll content in flag leaves. Higher TGW and chlorophyll content has been negatively regulated by *TaCKX2.1* expression as well, which in the case of TGW up-regulated contents of active cytokinins, iP and DZR, and glucosides of tZ. Moreover, TGW is positively regulated by *TaCKX9*, leading to increased tZR content. Interestingly, such an opposite effect to *TaCKX2.1* in *TaCKX1* silenced plants was maintained by *TaCKX11*, expressed in more developed spikes as well. Likewise, in the case of *TaCKX1* silencing [11], total CKX activity was not changed, but the proportions of specific isozymes are expected to be modified. These modifications of pulled isozymes might be responsible for highly increased cZ content. The opposite effect of *TaCKX2.1* on TGW and chlorophyll content in flag leaves was documented in our earlier report, in which silencing of *TaCKX1* resulted in lower TGW and SPAD values, which was regulated by increased expression of *TaCKX2.1* [11]. Therefore, we confirmed our previous results that *TaCKX2.1* in co-operation with other *TaCKX* GFMs downregulates TGW and with *TaCKX2.2.2* chlorophyll content in flag leaves. Similar correlations with grain size, weight and grain filling rate were reported for the allele of *TaCKX6a02* [35], which is *TaCKX2.1–3D* [12]. Opposed data were reported for other *TaCKX2* GFM, *TaCKX6* [36], renamed as *TaCKX2.2.1–3D* [12]. An 18-bp deletion in intron 2 of the gene resulted in its decreased expression, which was associated with a greater TGW, compared to its wild-type haplotype. However, these and our data are not well comparable because *TaCKX2.2.1* in our research is only slightly decreased and nature of these genetic modifications in both genomes are different. The results of Zhang et al. [36] refer specifically to the mutation of one homoeologous gene, and in our case all three homoeologous genes were slightly downregulated.

The changes observed by us in phytohormonal homeostasis are consistent with the obtained phenotype of higher TGW. Lack of GA and decreased ABA content in 7 DAP spikes of *TaCKX2.2.2* silenced plants should have resulted in delay of grain germination and effect on grain filling, which in consequence led to higher TGW and chlorophyll content in flag leaves. This result was proved by the opposite effect in *TaCKX1* silenced plants, showing ten times increased GA content. Moreover, growth of TGW in *TaCKX2.2.2* silenced plants was positively correlated with iP content, which was higher than in *TaCKX1* silenced plants. As reported, [56] decrease in the level of this phytohormone was associated with seed maturation in wheat. Therefore, increased iP is expected to delay this process in *TaCKX2.2.2* silenced plants, which in turn resulted in higher TGW. It was also documented that tZ and other CKs are implicated in the control of seed dormancy induction and maintenance, and IAA level is decreased during seed maturation [56,57]. As suggested by Fahy et al. [58], final grain weight is largely determined by developmental processes prior to grain filling.

The role of CKs in maintaining chlorophyll level in cereals has already been reported [8,59,60]. Down-regulation of *TaCKX2.1* and *TaCKX2.2.2* in 7 DAP spikes of *TaCKX2.2.2* silenced plants resulted in down-regulation of BA and ABA, lack of GA_3_ and up-regulation of free base isoprenoid cytokinins, cZ, tZ, iP. These changes positively influenced chlorophyll content in flag leaves of silenced plants. In contrast, up-regulation of *TaCKX2.1* in *TaCKX1* silenced plants was correlated with a slight increase of tZ, iP and cZ content and a very large increase of GA_3_, which resulted in lower chlorophyll content/early senescence of flag leaves [11]. It has already been documented that premature leaf senescence resulted in lower grain yield [59] and increased CK during seed development might improve chlorophyll content and determine grain number [61,62]. Moreover, ABA was reported to play an important role in leaf senescence as well [63,64]. The longer stay-green characteristic of flag leaves is expected to influence longer grain filling and higher TGW, since during further seed development nutrient compounds and phytohormones are transported from flag leaves to the seeds [38,60]. Similar results as for *TaCKX2.2.2* and *TaCKX1* silent plants of wheat were reported for knock-out *OsCKX11* in rice, which was significantly induced by ABA [38]. Increase of CK levels, mainly tZ and cZ was correlated with reduction in ABA levels determining delayed leaf senescence and increased branch, tiller, and grain number. Therefore, both *TaCKX* genes and *OsCKX11* function antagonistically between CKs and ABA in leaf senescence of wheat and rice.

Decreased *TaCKX2.2.2* expression resulted in higher TGW but reduced grain number. The same gene also participates in regulation of spike length and root weight and, in the opposite way, in chlorophyll content of flag leaves. The gene is strongly, positively correlated with ABA, involved in regulation of all these traits. As would be expected, grain number, grain yield, and spike number are oppositely regulated to chlorophyll content by BA and ABA. Moreover, cZOG, negatively correlated with *TaCKX2.2.2*, is involved in strong, negative regulation of grain number and spike length but positively with spike number. The latter trait is also importantly regulated by BA and ABA as well as cZ. Therefore, the spike number is up-regulated by cZ and down-regulated by BA, which is co-ordinated by ABA. The content of ABA in seven DAP spikes, which is the milk/early dough stage of wild-type wheat plants was low, about 2.6 ng/g biomass [11] and decreased by 50% in *TaCKX2.2.2* silenced plants. Since the highest concentrations of ABA were detected in the late dough stage [13], we may suppose that the lower level of ABA in seven DAP spikes resulted in extending chlorophyll content in flag leaves and also the maturation time of the spikes, which led to higher TGW. High content of cZ was observed during early embryo development of barley, underlying its important role in this stage of development [57]. cZ has also been detected in other cereal crop species such as rice [65] and maize [66], but its role in spike development was not known. Similarly, as in maize, the *c*Z of wheat was preferentially glycosylated to *O*-glucoside. The content of this conjugate was negatively correlated with grain number, grain yield, spike length and plant height. Since the content of cZOG was not changed, and only grain number was reduced, it could be positively affected by decreasing BA. A positive correlation of BA with grain number, grain yield, and spike number and negative with chlorophyll content underline its role in posttranscriptional or posttranslational regulation of protein abundance involved in ABA biosynthesis and response [67]. Again, we are not able to confirm that silencing by RNAi of expression of *TaCKX2.2.1–3A* (originally *TaCKX2.4*) determines grain number per spike [34]. Such a differences might be dependent on individual genetic background, which modulates co-operation of other *TaCKX* GFMs or other regulatory genes, existing in that background, which take part in establishing hormonal homeostasis.

*TaCKX9* is one more gene which coordinates development of TGW as well as plant height and grain yield in *TaCKX2.2.2* silenced plants. The gene is expressed in seedling roots as well as younger developing organs, showing very high expression in leaves [32], and is co-expressed positively with similarly expressed *TaCKX5* [33]. We revealed a positive correlation of *TaCKX9* expression with the riboside of tZ. Since tZR belongs to transporter metabolites [20], and the gene is expressed through the developing plant, we might predict its role in coordination of other genes involved in transport of tZR from roots to developing spikes. Indeed, there is a positive correlation between root weight and tZR and a negative correlation between plant height/TGW/grain yield and tZR measured in seven DAP spikes.

## 4. Materials and Methods

### 4.1. Plant Material

The spring wheat (*Triticum aestivum* L.) cultivar Kontesa, which is susceptible to *Agrobacterium*-mediated transformation and in vitro regeneration, was used in all experiments.

### 4.2. Vector Construction

The binary vector pBract207 (https://www.jic.ac.uk/technologies/crop-transformation-bract/, accessed on 10 March 2021) containing the hpRNA type of silencing cassette was assembled via Gateway cloning. In the first step, the 385 bp fragment of a coding sequence of *TaCKX2.2.2–3D* (FJ648070, 1259., 1643; TraesCS3D02G143300) was amplified using the primers: forward 5′- CTGGCTCAACCTGTTCCTC-3′, reverse 5′- ATACTTCCTCTTCCGATCCACG-3′ and cloned into the entry vector pCR8/GW/TOPO (Invitrogen, Karlsbad, CA, USA). This fragment has 93%–94.3% identity with the same fragments of *TaCKX2.2.1*, located on 3A, 3B, and 3D chromosomes and 87.9%–92.3% identity with *TaCKX2.1*, located on 3A, 3B, and 3D chromosomes as well (Appendix A). Next, the *TaCKX2.2.2* insert from the entry vector was cloned to the pBract207 destination vector in a sense and antisense orientation using LR Clonase (Invitrogen) according to the manufacturer’s protocol. The correct assembly of the silencing cassette in the pBract207 vector was verified by restriction enzyme analysis and Sanger sequencing. The vector was then electroporated into the AGL1 strain of *Agrobacterium tumefaciens* and used for transformation.

### 4.3. Plant Transformation and Selection of Transgenic Lines

Donor plants for transformation experiments were grown in growth chambers under controlled environmental conditions at 20 °C day and 18 °C night temperatures, and a 16 h light/8 h dark photoperiod. Light intensity of 350 μmol/m^2^/s was provided by fluorescent lamps. *Agrobacterium*-mediated transformation of immature embryos and in vitro plant regeneration were performed according to our previously developed protocols for wheat [68,69]. Putative transgenic events as well as T_1_ plants were verified by PCR screening for the presence of a T-DNA fragment using primers specific to the *hpt* gene. PCR screening was performed for T_0_ plants as well as 14-day old T_1_ seedlings using the KAPA3G Plant PCR Kit (Kapa Biosystems, Wilmington, MA, USA). The following primers were used for the amplification of a 405 bp fragment of the *hpt* gene: forward 5′-ATGACGCACAATCCCACTATCCT-3′ and reverse 5′-AGTTCGGTTTCAGGCAGGTCTT-3′. The reaction was carried out in a final volume of 50 µl containing 1X KAPA Plant PCR Buffer, 0.3 μM of each primer, and 1 U of KAPA3G Plant DNA Polymerase. A 0.5 × 0.5 mm fragment of leaf tissue was used as a template. The PCR was run with the following temperature profile: an initial denaturation step at 95 °C for 3 min.; 40 cycles of 95 °C for 20 s, 68 °C for 30 s, 72 °C for 30 s, and the final extension step at 72 °C for 2 min. The PCR products were separated and visualized on agarose gels. Respective generation of control, in-vitro plants was applied in verification by PCR screening. Non-transgenic null segregants were used as control samples in all experiments.

### 4.4. Quantitative RT-qPCR

Total RNA from was extracted from the middle part of 7 DAP (days after pollination) spikes using TRI Reagent (Sigma-Aldrich, St. Louis, MO, USA) according to the manufacturer’s protocol. Purification of RNA and cDNA synthesis was performed as described by Ogonowska et al. [32]. RT-qPCR analysis was done for seven target genes: *TaCKX1* (JN128583), *TaCKX2.1* (JF293079), *TaCKX2.2.1* (FJ648070), *TaCKX2.2.2* (GU084177), *TaCKX11* (former *TaCKX3*) (JN128587), *TaCKX5* [32] and *TaCKX9* (former *TaCKX10*) (JN128591). Sequences of specific primers designed for the genes tested are shown in Appendix A. ADP-ribosylation factor (*Ref2*) (AB050957) was used as a reference gene in all qPCR reactions. qPCR was carried out in a 10-µl mixture containing 1x HOT FIREPol EvaGreen qPCR Mix Plus (no ROX) (Solis BioDyne, Tartu, Estonia), 0.2 µM of each primer and 4 µL of 20 x diluted cDNA. The following temperature profile was used: an initial denaturation/polymerase activation step of 95 °C for 12 min; 45 cycles of amplification at 95 °C for 20 s, 62 °C for 20 s, 72 °C for 20 s; final denaturation step at 72 °C for 5 min and the melting curve profile from 72 °C to 99 °C, with the temperature rising 1 °C at each step and continuous fluorescence measurements. The expression levels of *TaCKX* GFMs were calculated from three technical replicates (repeated samples in RT-qPCR reaction) according to the standard curve method using the *Ref2* gene as a normalizer [70]. The relative expression levels in transgenic/silenced plants were calculated as x-fold of 1 based on mean expression value in null segregants, which was set to 1. The analysis was performed for three biological replicates.

Experimental plants were divided into three groups depending on the level of silencing: ≤0.60 as highly silenced, 0.61–0.79 as silenced and ≥0.80 as non-silenced. This three-group scale is used to find if there was dependence between expression level of silenced genes and phenotype.

### 4.5. CKX Activity Assays

Cytokinin oxidase/dehydrogenase activity was measured in 7 DAP spikes of T_1_/T_2_ plants as described previously [9,33].

### 4.6. Quantification of ABA, Auxins, Cytokinins and GA_3_

For quantification, the standards of phytohormones were used: ABA, five auxins: indole-3-acetic acid (IAA), indole-3-butyric acid (IBA), indole-3-propionic acid (IPA), 1-naphthaleneacetic acid (NAA), and 2-phenylacetic acid (PAA); twenty-seven standards of CKs: *trans*-zeatin (tZ), *trans*-zeatin riboside (tZR), *trans*-zeatin-9-glucoside (tZ9G), *trans*-zeatin-7-glucoside (tZ7G), *trans*-zeatin-O-glucoside (tZOG), *trans*-zeatin riboside-O-glucoside (tZROG), *trans*-zeatin-9-glucoside-O-glucoside (tZ9GOG), *trans*-zeatin-9-glucoside riboside (tZ9GR), cZ, *cis*-zeatin-riboside (cZR), *cis*-zeatin O-glucoside (cZOG), *cis*-zeatin 9-glucoside (cZ9G), *cis*-zeatin-O-glucoside-riboside (cZROG), dihydrozeatin (DZ), dihydrozeatin-riboside (DZR), dihydrozeatin-9-glucoside (DZ9G), dihydrozeatin-7-glucoside (DZ7G), dihydrozeatin-O-glucoside (DZOG), dihydrozeatin riboside-O-glucoside (DZROG), N^6^-(Δ^2^-isopentenyl)adenine (iP), N^6^-isopentenyladenosine (iPR), N^6^-isopentenyladenosine-7-glucoside (iP7G), *para*-topolin (pT), *meta*-topolin (mT), *ortho*-topolin (oT), 6-benzylaminopurine (6-BAP), and GA_3_.

The procedure of phytohormone quantification is described in Jablonski et al. [11].

### 4.7. Analysis of Phenotypic Traits

The following parameters were determined after harvesting mature plants: plant height, spike length, number of spikes per plant, number of grains per plant, weight of grains per plant and 1000 grain weight. Additionally, the weight of fresh mass of roots from 5-day old seedlings was determined in T_2_ lines. Seeds were germinated on wet glass beads in Petri dishes for 5 days in the dark, at room temperature. The excess of water in the roots was removed by placing them on the filter paper and then, the roots were immediately cut from the seedlings and weighed. Chlorophyll content was measured in the flag leaves from the first five tillers with spikes, periodically from 0 DAP up to leaf desiccation using a SPAD chlorophyll meter (Konica Minolta SPAD-502Plus). All measurements were performed for six biological replicates.

### 4.8. Statistical Analysis

All statistical analyses were performed using Statistica v13.3 software (StatSoft, Kraków, Poland) and the mean ± SE was presented. The normality of the tested samples was verified by the Shapiro–Wilk test. Differences between tested and control samples in phytohormone content, phenotypic traits and expression levels were verified by Student’s t-test or the Mann–Whitney test at *p* < 0.01 and *p* < 0.05 confidence levels. Correlation coefficients of analyzed traits were verified by either the parametric Pearson test or the nonparametric Spearman test.

## 5. Conclusions

We documented that silencing of one of the *TaCKX* GFMs co-ordinates the expression of other genes from the family. This leads to a homeostatic feedback loop between the transcription of *TaCKX* genes and phytohormones and establishing phytohormonal homeostasis. Tested *TaCKX* genes showed specific correlations to the forms/metabolites of CKs and are involved in the regulation of other phytohormones. The *TaCKX2.2* genes strongly modulate yield-related traits of wheat by cross-talk with other *TaCKX* GFMs and establishing homeostasis of phytohormones in developing wheat plants. 

## Figures and Tables

**Figure 1 ijms-22-04142-f001:**
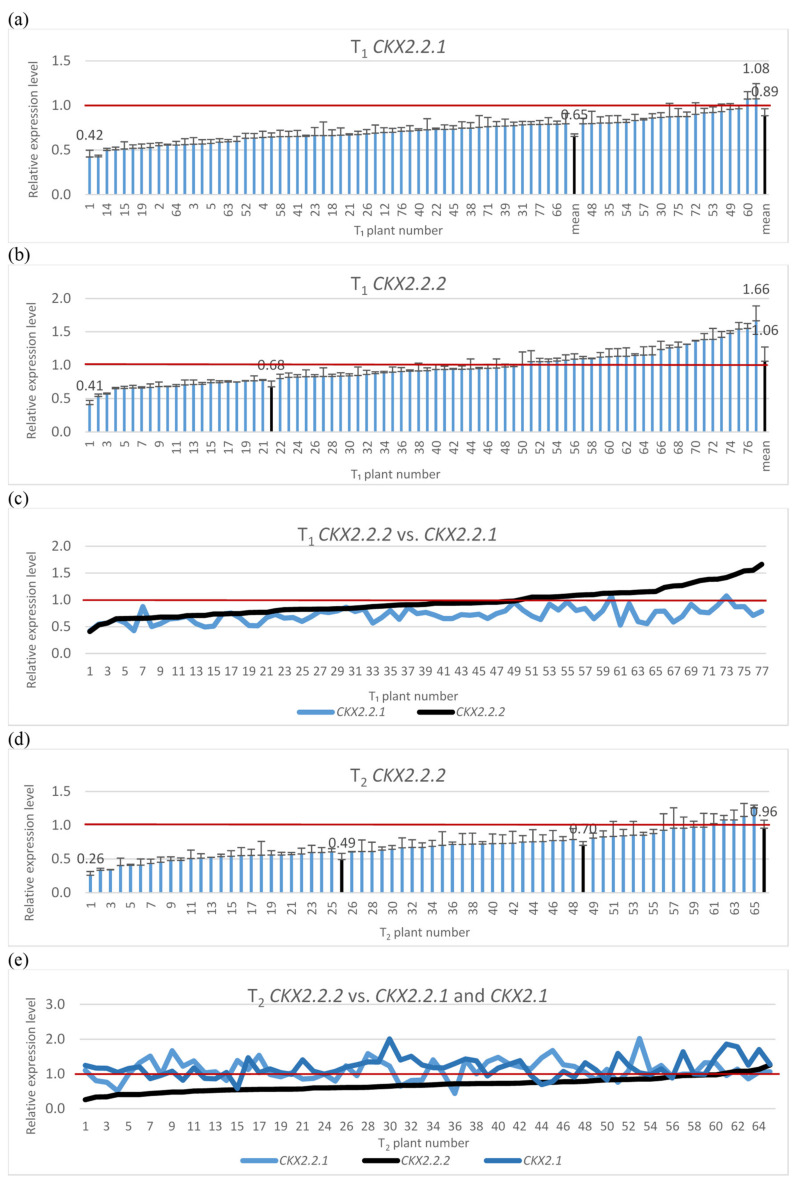
Relative expression levels of silenced *TaCKX2.2.1* (**a**), *TaCKX2.2.2* (**b**), *TaCKX2.2.2* vs. *TaCKX2.2.1* (**c**) in T_1_ and *TaCKX2.2.2* (**d**), and *TaCKX2.2.2* vs. *TaCKX2.2.1* and *TaCKX2.1* (**e**). The error bars denote ± SD. The thresholds of classified plants are 0.6 and 0.8.

**Figure 2 ijms-22-04142-f002:**
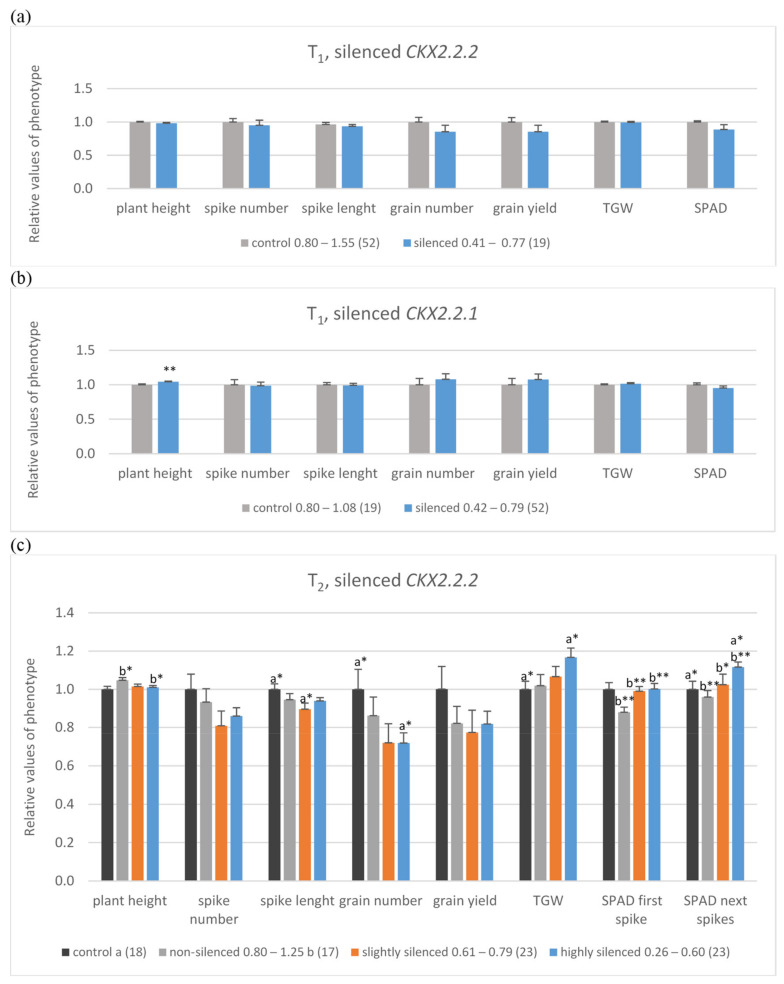
Mean, relative values of yield-related traits in control, non-silenced (grey bars), silenced (orange bars) or highly silenced (blue bars) plants of T_1_ with silenced *TaCKX2.2.2* (**a**) and *TaCKX2.2.1* genes (**b**) and T_2_ lines with silenced *TaCKX2.2.2*, including R_2_, control (black bars) (**c**). SPAD was measured on the flag leaves of the first (SPAD first spike) or four next spikes (SPAD next spikes). The error bars denote ± SE. Different letters mean significant differences: *—significant at *p* ≤ 0.05; **—significant at *p* ≤ 0.01.

**Figure 3 ijms-22-04142-f003:**
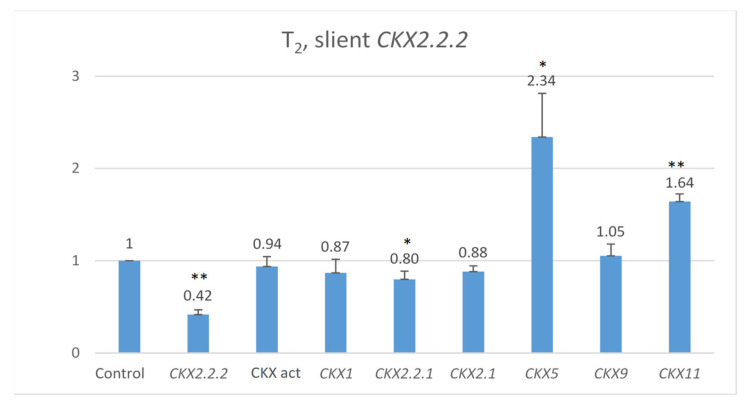
Relative expression of other *TaCKX* GFMs in 7 DAP spikes, related to the control, non-silent *TaCKX2.2.2* set as 1.00, and relative CKX enzyme activity (CKX act bar). The error bars denote ± SE. *—significant at *p* ≤ 0.05; **—significant at *p* ≤ 0.01.

**Figure 4 ijms-22-04142-f004:**
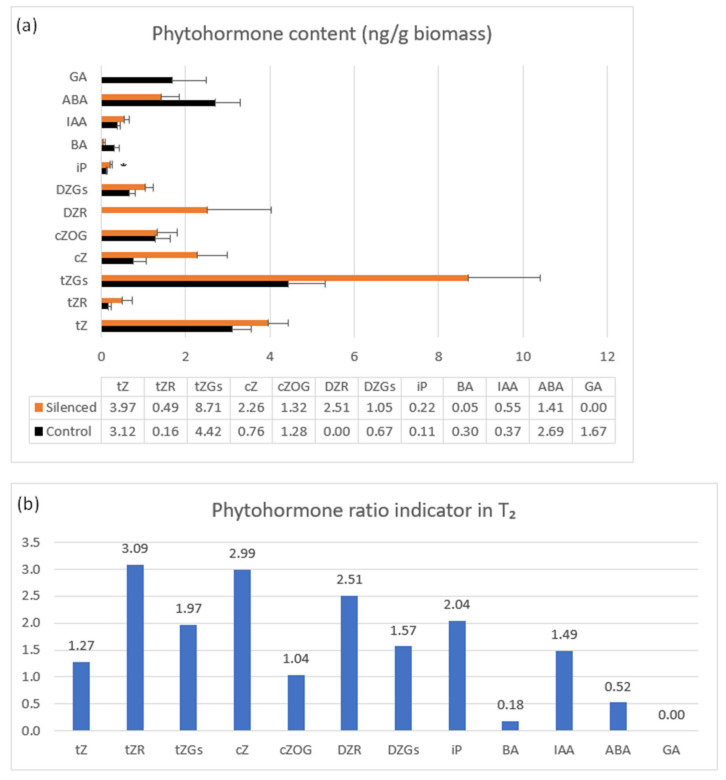
Phytohormone content in control (black bars) and *TaCKX2.2.2* silent T_2_ plants (**a**) and phytohormone ratio indicator (**b**). Not shown: small amounts (≤0.20 ng/g biomass): cZOGR, DZ7G, DZOGR, iP, iP7G, BA; trace amounts (≤0.02 ng/g biomass): cZR, cZ9G, DZ9G. Not detected: DZ, iPR, IBA, IPA, PAA. The error bars denote ± SE. * *p* ≤ 0.05.

**Figure 5 ijms-22-04142-f005:**
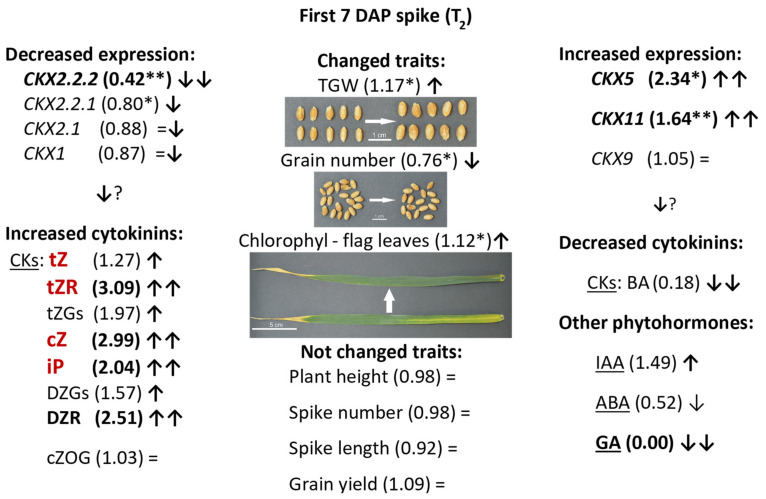
Schematic presentation of up (↑) or down (↓) regulation of expression of *TaCKX* GFMs, phytohormone content and yield-related traits in 7 DAP spikes of silenced, T_2_ lines. Relative to the control = 1.00 values are in brackets. Highly decreased (↓↓) or highly increased (↑↑) values are in bold. *—significant at *p* ≤ 0.05; **—significant at *p* ≤ 0.01. CKs in red are substrates for CKX enzyme. =—not changed; ?—expected results.

**Figure 6 ijms-22-04142-f006:**
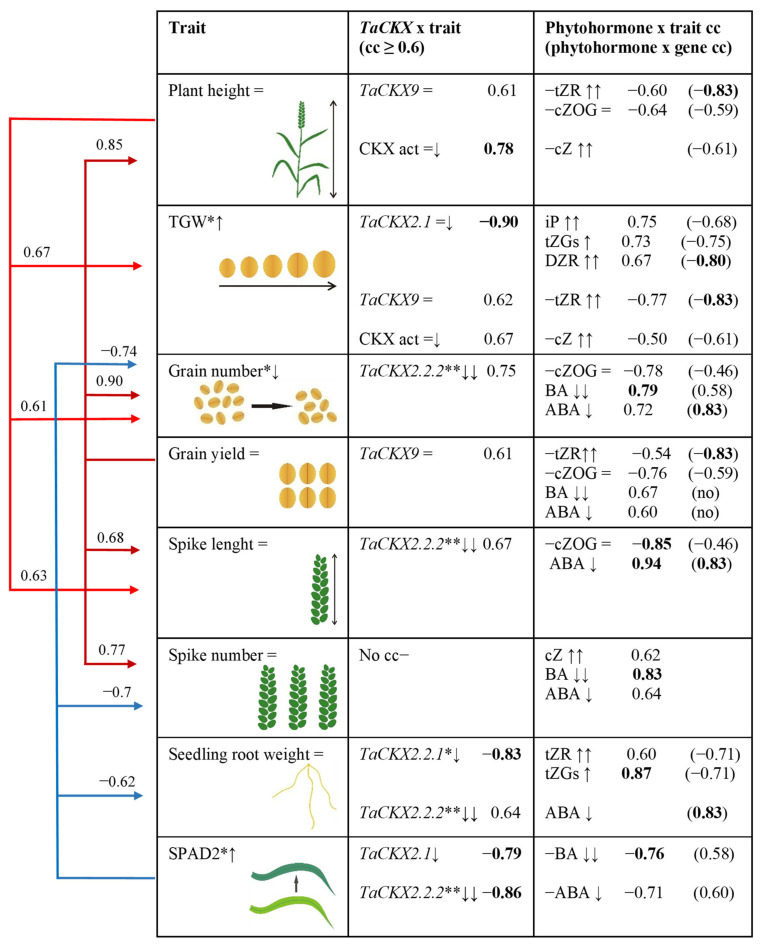
Effect of increased (↑), decreased (↓) or unchanged (=) expression of *TaCKX* GFMs as well as phytohormones in 7 DAP spikes on yield-related traits based on correlation coefficients (cc). Red arrows—positive cc; blue arrows—negative cc. Values of cc in bold are significant at *p* < 0.05; no—no correlation; =↓ slightly silenced; ↓↓ highly silenced. *—significant at *p* ≤ 0.05; **—significant at *p* ≤ 0.01.

**Table 1 ijms-22-04142-t001:** Correlation coefficients between *TaCKX* GFMs and phytohormones. Values in bold are significant at *p* < 0.05. = not changed; =↓ slightly silenced; ↓ silenced or down-regulated; ↓↓ highly silenced, significant; ↑ increased expression or up-regulated phytohormone.

*TaCKX*	cc Phytohormones
*TaCKX1* =↓	↓tZR −0.66 ↓AA −0.71
*TaCKX2.1* =↓	↓tZGs −0.75 ↓DZR −**0.80** ↓iP −0.68
*TaCKX2.2.1* ↓	↓tZR −0.71 ↓tZGs −0.71 ↓IAA −0.77
*TaCKX2.2.2* ↓↓	↑BA 0.58 ↑ABA **0.83**
*TaCKX5* ↑↑	↓tZR −0.77 ↓IAA −**0.83**
*TaCKX11* ↑↑	↓ABA −0.77
*TaCKX9* =	↓tZR −**0.83** ↓cZOG −0.59 ↓IAA −0.54

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
