# Peer review of "TaCKX2.2 Genes Coordinate Expression of Other TaCKX Family Members, Regulate Phytohormone Content and Yield-Related Traits of Wheat"

_ijms, 2021, doi:10.3390/ijms22084142_

Round 1

Reviewer 1 Report

In this manuscript, the authors utilized RNAi technology to knock down the expression of TaCKX genes by one RNAi fragment, and then analyzed their contribution to yield-related traits, chlorophyll content and phytohormones. However, the data and analysis were not presented well in this manuscript.

Due to the defect of off-targets, RNAi technology is not good method for demonstrating the special functions of one or two members of gene family as shown in this manuscript. Actually, the authors use one fragment for silencing three genes TaCKX2.2.1, TaCKX2.2.2 and TaCKX2.1 as shown in methods. Although the silencing efficiency of TaCKX2.1 is lower than silencing efficiency of TaCKX2.2.1 and TaCKX2.2.2, the authors can’t ignore the knocking down of TaCKX2.1 when analyze their data. Additionally, due to the different roles of different genes, the knocking-down level for each gene can’t determine or distinguish their functional importance in plants. Here, I can only see the co-silencing of TaCKX2.2.1, TaCKX2.2.2 and TaCKX2.1 (maybe off-target other TaCKX gene members, such as CKX1) lead to the changes of phytohormone content and yield-related traits of wheat. So the authors need to re-organize the data and manuscript.

Since there are 13 TaCKX GFMs, why the authors only detected seven target genes: TaCKX1 (JN128583), TaCKX2.1 542 (JF293079), TaCKX2.2.1 (FJ648070), TaCKX2.2.2 (GU084177), TaCKX11 (former TaCKX3) (JN128587), TaCKX5 and TaCKX9 (former TaCKX10) (JN128591). The authors didn’t show the analysis of off-targets by RNAi. Whether the 385 bp fragment for RNAi can target other TaCKX genes or affect the expression of other TaCKX genes as shown in figure 3 CKX1? Moreover, the authors need to show the sequence alignments of RNAi fragment with TaCKXs genes.

Line 143, the description here was not accurate. The silencing cassette targets three genes TaCKX2.2.1, TaCKX2.2.2 and TaCKX2.1.

Figure 1, the authors should also analyze the expression of CKX2.1 which is the direct target of RNAi fragment in T1 plants. The RNAi fragment has 91.3 to 92.4 % identity with TaCKX2.1 showed by the authors. The expression of CKX2.1 in T1 plants may contribute the Yield-related traits as analyzed in results 2.2, line 162-174.

Line 595, actually, the authors use one fragment for silencing of three genes TaCKX2.2.1, TaCKX2.2.2 and TaCKX2.1 as shown in Materials and Methods 4.2. The description here was not accurate.

Author Response

Dear Reviewer (1),

Thank you very much for your effort to review our manuscript. Our responses and explanations are under your queries. We believe that you find them acceptable. Please note that all Figures were corrected and replaced with previous ones.

Best regards,

Anna Nadolska-Orczyk

“Due to the defect of off-targets, RNAi technology is not good method for demonstrating the special functions of one or two members of gene family as shown in this manuscript. Actually, the authors use one fragment for silencing three genes TaCKX2.2.1, TaCKX2.2.2 and TaCKX2.1 as shown in methods. Although the silencing efficiency of TaCKX2.1 is lower than silencing efficiency of TaCKX2.2.1 and TaCKX2.2.2, the authors can’t ignore the knocking down of TaCKX2.1 when analyze their data. Additionally, due to the different roles of different genes, the knocking-down level for each gene can’t determine or distinguish their functional importance in plants. Here, I can only see the co-silencing of TaCKX2.2.1, TaCKX2.2.2 and TaCKX2.1 (maybe off-target other TaCKX gene members, such as CKX1) lead to the changes of phytohormone content and yield-related traits of wheat. So the authors need to re-organize the data and manuscript.”

A.N.-O.: In the original manuscript we considered all the points raised by the Reviewer (1). Discussing the results we took into consideration several important points i) the levels of silencing of particular genes, ii) the influence of this silencing on the coregulation of other GFMs, iii) the phytohormone homeostasis and iv) the regulation of yield-related traits. We did not ignore the fact that the silencing fragment in the RNAi cassette could induce silencing of the TaCKX2.1. In response to this suggestion, we added Table S4 to Supplementary Materials. The effect of silencing, which is directly related to the expression level of the silencing cassette, has been measured by RT-qPCR. All results are discussed considering the components of this complex system. We argue, and we discuss this point in the manuscript, that i) the coregulation of TaCKX genes, ii) the differences in their expression in plants of T1 and T2 generation combined with ii) the data on phytohormone levels as well as iv) the phenotypes create a very good basis to draw the conclusions presented in the manuscript. The differences of individual gene silencing found within one generation and the differences between generations were correlated with the phenotypes. In our opinion, this set of data additionally supports the conclusions. The general suggestion to re-organize data in the manuscript is not supported by anything specific and we would like to point that none of the remaining Reviewers raised such comments.

“Since there are 13 TaCKX GFMs, why the authors only detected seven target genes: TaCKX1 (JN128583), TaCKX2.1 542 (JF293079), TaCKX2.2.1 (FJ648070), TaCKX2.2.2 (GU084177), TaCKX11 (former TaCKX3) (JN128587), TaCKX5 and TaCKX9 (former TaCKX10) (JN128591). The authors didn’t show the analysis of off-targets by RNAi. Whether the 385 bp fragment for RNAi can target other TaCKX genes or affect the expression of other TaCKX genes as shown in figure 3 CKX1? Moreover, the authors need to show the sequence alignments of RNAi fragment with TaCKXs genes.”

A.N.-O.: From our introduction: “According to the latest revision by Chen et al. [12] there are 13 TaCKX GFMs basically numbered from 1 to 11, and 12 of them are located on the three homoeologous chromosomes of the A, B and D genome, giving the number of 35 GFMs.” For our analysis we chose selected GFMs specifically expressed in three groups of selected organs (indicated in introduction and discussed). The manuscript presents huge set of data collected during 4-year long experiments and, as it was discuused they support the conclussions. The analysis of off-targets is done in the additional Table S4 of Supporting Materials.

“Line 143, the description here was not accurate. The silencing cassette targets three genes TaCKX2.2.1, TaCKX2.2.2 and TaCKX2.1.”

A.N.-O.: Thank you. This was corrected.

“Figure 1, the authors should also analyze the expression of CKX2.1 which is the direct target of RNAi fragment in T1 plants. The RNAi fragment has 91.3 to 92.4 % identity with TaCKX2.1 showed by the authors. The expression of CKX2.1 in T1 plants may contribute the Yield-related traits as analyzed in results 2.2, line 162-174.”

A.N.-O.: The T1 generation was tested three years ago and, at this time there was no information about the whole GFMs. The first, full classification has been done by Chen et al. 2020. We wanted to complete the data, however the RNA samples from the T1 plants were already 3-years old, therefore the analysis of TaCKX2.1 was done only using plants of T2 generation. However in our opinion, the lack of the analysis of the T1 plants did not affect either the correlation analyzes or the conclusions.

“Line 595, actually, the authors use one fragment for silencing of three genes TaCKX2.2.1, TaCKX2.2.2 and TaCKX2.1 as shown in Materials and Methods 4.2. The description here was not accurate.”

A.N.-O.: This description is about the effect of highly silenced TaCKX2.2.2 found  in the T2 generation (these results are presented in all Figures and underlined in Figure 5 and Figure 6). Such coordinated effect of one gene silencing on up- or down-regulation of other genes has been already documented in our earlier paper on TaCKX1 silencing (Jablonski et al. 2020, position 11 in the references).

Best regards,

Anna Nadolska-Orczyk

Reviewer 2 Report

The manuscript “TaCKX2.2 genes coordinate expression of other TaCKX family 2 members, regulate phytohormone content and yield-related 3 traits of wheat” by Jablonski et al. is a well-designed, good elaborated and interesting study that concerns rather relevant problem of the impact of phytohormone-regulating genes on yield-related traits. However, there certain points that should be corrected or addressed. In general, the demonstration of the results should be more consistent and the discussion should be more clearly organized.

Here is the list of questions and comments:

Abstract

Comment: Please, specify in abstract, in which organs the gene expression phytohormone content were measured.

Introduction

line 53 “root (and shoot) growth”

Comment: why did you use parentheses? are they necessary here?

line 67 “N6-(Δ2-isopentenyl)adenine”

Comment: in line 572 you use N6-isopentenyladenine (iP). What is the difference? Are they the same?

line 68 “cis-zeatin (cZ)”

Comment: cis- should be in italics (latin)

line 87 “traits in cereals, inter alia rice”

Comment: inter alia should be in italics (latin)

line 112 “effect of TaCKX2.4 gene silencing by RNAi was increased grain number per spike”

Comment: “was” should be removed

Results

Line 142 Section 2.1. Expression levels of silenced TaCKX2.2.1, TaCKX2.2.2 and TaCKX2.1 in T1 and T2

Comment: Judging by the description of the results and Table S1 you use the following scale:

<0.60 highly silenced

0.61-0.79 silenced

>0.80 non-silenced

Please, include the scale in Material and methods. Please, give the reference to you other studies where this scale is used or explain the you used this specific scale in a given study for specific reasons.

line 148-149 “Silencing levels”

Comment: What is the difference between “silencing level” and “expression level”? Would not it be more easily for a reader if you use the terms “expression level” term and “silencing”? Relative expression level is measured as the expression level of the silenced plant related to a control, non-silenced plant. What is then “silencing level”? As to me, this term impedes the comprehension of the results and just “silencing” should be used, instead.

line 151 “(N 79)”

Comment: This should be deciphered. If I understood it correctly, it designates the sample volume. Please, specify what “N” means here.

Line 170 “TGW and SPAD”

Comment: Please, decipher these abbreviations here and, for SPAD, in Material and methods section.

It is a good idea to give the whole list of the abbreviations with their meaning at the beginning of the manuscript or in Supplemental materials.

Lines 172 “Table S1 a”

Comment: Delete space between S1 and a.

Lines 174 “slightly higher compared to the control”

Comment: Here, you should specify what was used as a control (non-transgenic null segregants, according to line 537)

Line 175 “Among 65 T2 analyzed plants”

Comment: 2 in T2 should be in lowercase.

Line 177-178 “Additionally 18 plants were not- silenced”

Comment: You need comma after “Additionally” and no space between “not-” and “silenced,’

Figure 1.

Comment: Generally, the top of each chart should be in the middle of the bars. In you diagram the bars look strange as if they are on the top of the column. Would you kindly explain why?

Please, put the letter (a, b, c, …,e) in the high left corner of each diagram as it would be more convenient to find it wheat one is reading the figure.

Please, specify what is shown in x-axis.

It would be a good idea to show the thresholds (0.6 and 0.8) that you used to classify the produced plants in the diagrams.

Line 188 “and TaCKX2.2.2 vs. TaCKX2.2.1

Comment: according to the diagram, it should be “and CKX2.2.2 vs. CKX2.2.1 and CKX2.1 (e)”

Here, the general comment: in some places you use TaCKX, in others, CKX. For example, in Figure 1 there is inconsistencies in gene designation between Figure caption and the diagram names in prefix “Ta-“. In sections 2.1 and 2.2 you describe the geens as CKXs, while in 2.3 and 2.4 as TaCKXs. Please, make it uniform throughout the manuscript where CKXs are described specifically as wheat genes.

Lines 196-206

Comment: Here, the TaCKXs genes should be given in italics. And again, please, use TaCKX (not CKX where wheat genes are considered) designation throughout the whole manuscript as different titles hamper understanding the text.

Figure 3

Comment: What is shown in “CKX act” chart? Please, decipher in Figure caption and add the description of this column in the respective abstract.

Line 223-224 “trace amounts (≤ 0.02 ng/g biomass…, DZ, …. Not detected: DZ,…”

Comment: Here is inconsistence: was DZ not detected or detected in trace amounts?

Figure 5.

Comment: It is a good idea to use symbols to designate the relative expression level such as = (not changed significantly), =↓ (slightly silenced, not significant), ↓ (silenced, significant), ↓↓ (highly silenced, significant), ↑ (expression increased). So, please, use them uniformly in Figure 3, Figure 5, Table 1, Figure 6. In Figure 3 you may provide the explanation of these symbols in statistical language (significant/ not significant, p<0.05) and hereafter use them in following tables and figures. In particular, replace “nc” by “=” in Figure 5.

Lines 230-231

Comment: The text is disrupted here. Please, correct the text makeup

Table 1

Comment: I suggest that you use up and down arrows instead of “-“ in “cc phytohormones” column. It will demonstrate the negative and positive feedback better.

In TaCKX2.2.2↓↓ you use parentheses for (BA 0.58) and exclamation mark for ABA 0.83! Would you please explain why.

Figure 6 “-CKX2.1”, “-CKX2.2.1↓”, “-CKX2.1”

Comment: What does “-” (minus) mean here? Since it decreased you do not need minus. It really mislead and should be explained.

Line 305 Discussion

Comment:Here, at the beginning of Discussion, as well as in the end of Introduction you should briefly mention that you in your previous work you silenced TaCKX1 and this is (to certain extent) a continuation of that research. In this context, the discussion will be more easily comprehended. You here and then reference to your research [11] without mentioning that it is your study. I find it correct to reference to it openly, as it makes the results of [11] and the present study comparable and clear to understand.

Line 363. 3.3. Homeostatic feedback loop between TaCKX genes transcription and phytohormones

Comment: The idea of feedback loop is great. I suggest that you provide a scheme/model with arrows that demonstrate these feedback loops that proved homeostatic state in plant. For example: TaCKX2.2.2, TaCKX2.2.1, TaCKX1 down -> cZ and iP up-> TaCKX5 up->IAA down. You may develop this scheme for plant based on your and literature results as it may really contribute to the whole understanding of such links between genes-hormones nets.

Lines 406-441, lines 442-489

Comment: the paragraphs are too large. I suggest that you divide them into several paragraphs according to the main ideas in them.

Line 420 “down regulates”

Comment: the space is excess, “downregulates”

Lines 421-428: “Our data are not in agreement with those reported for TaCKX6 [36], renamed as TaCKX2.2.1-3D [12]. An 18-bp deletion in intron 2 of the gene resulted in its decreased expression, which was associated with a greater TGW, compared to its wild-type haplotype. However, these and our data are not well compara-ble because of different nature of these genetic modifications in the genomes. The results of Zhang et al. [36] refer specifically to the mutation of one homoeologous gene, and in 427 our case all three homoeologous genes were downregulated.”

Comment: In sentences before and after this fragment you discuss TaCKX2.2.2 and TaCKX1. But here you discuss TaCKX2.2.1. It confuses a little and makes it hard to follow the discussion. Moreover, you write that “Our data are not in agreement”. But in your experiments TGW slightly (insignificantly) decreased in T1 TaCKX2.2.1-silinced plants and in T2 you TaCKX2.2.1 was silenced along with TaCKX2.2.2 and TGW increased. What particular results are not in agreement? Please, specify.

Line 453 “longer grain felling”

Comment: “longer grain filling” should be instead

Line 546 vs line 554 Ref2/ref2

Comment: Please, use the same designation lowercase or uppercase letter.

Line 538 4.4 4.4. Quantitative RT-qPCR

Comment: What was the technical replicate?

What was the PCR efficiency? Please, demonstrate the calibration curves for the used primers.

Line 536-537 “Non-transgenic, null segregants were used as control samples in all experiments.’

Comment: How many plants were used as a control?

Line 565 “tZ”

Comment: Please, decipher (other substances are deciphered here)

References

Line 671 “Arabidopsis thaliana” should be in italics

Line 730 “Triticum-Aestivum” should be “Triticum aestivum” in italics

Supplementary Table S2

Comment: I cannot catch the idea of grey highlighting and asterisks (*) in the heads of the table. Would you please specify this?

The smanuscript can be recommended for publication after addressing the abovelisted notes and comments.

Kind regards,

Reviewer 

Author Response

Dear Reviewer (2),

Thank you very much for your very helpful and meticulous review, which improved the manuscript. We made point-by-point corrections in the manuscript and our responses are under your comments. Please note that all Figures were corrected and replaced with previous ones.

Best regards,

Anna Nadolska-Orczyk

Abstract. Comment: Please, specify in abstract, in which organs the gene expression phytohormone content were measured.

A.N.-O.: Thank you. “7 DAP spikes” are added.

Introduction

line 53 “root (and shoot) growth”

Comment: why did you use parentheses? are they necessary here?

A.N.-O.: Thank you. Parentheses are removed.

line 67 “N6-(Δ2-isopentenyl)adenine”

Comment: in line 572 you use N6-isopentenyladenine (iP). What is the difference? Are they the same?

A.N.-O.: They are the same. Corrected in line 572.

line 68 “cis-zeatin (cZ)”

Comment: cis- should be in italics (latin)

A.N.-O.: Thank you. Corrected.

line 87 “traits in cereals, inter alia rice”

Comment: inter alia should be in italics (latin)

A.N.-O.: Thank you. Corrected.

line 112 “effect of TaCKX2.4 gene silencing by RNAi was increased grain number per spike”

Comment: “was” should be removed

A.N.-O.: Thank you. Corrected.

Results

Line 142 Section 2.1. Expression levels of silenced TaCKX2.2.1, TaCKX2.2.2 and TaCKX2.1 in T1 and T2

Comment: Judging by the description of the results and Table S1 you use the following scale:

<0.60 highly silenced

0.61-0.79 silenced

>0.80 non-silenced

Please, include the scale in Material and methods. Please, give the reference to you other studies where this scale is used or explain the you used this specific scale in a given study for specific reasons.

A.N.-O.: The scale and explanation is included in Materials and Methods.

line 148-149 “Silencing levels”

Comment: What is the difference between “silencing level” and “expression level”? Would not it be more easily for a reader if you use the terms “expression level” term and “silencing”? Relative expression level is measured as the expression level of the silenced plant related to a control, non-silenced plant. What is then “silencing level”? As to me, this term impedes the comprehension of the results and just “silencing” should be used, instead.

A.N.-O.: Thank you. This is valuable comment, which is applied through the manuscript.

line 151 “(N 79)”

Comment: This should be deciphered. If I understood it correctly, it designates the sample volume. Please, specify what “N” means here.

A.N.-O.: Yes, it means number of plants. Explained.

Line 170 “TGW and SPAD”

Comment: Please, decipher these abbreviations here and, for SPAD, in Material and methods section.

A.N.-O.: Thank you. Corrected.

It is a good idea to give the whole list of the abbreviations with their meaning at the beginning of the manuscript or in Supplemental materials.

Lines 172 “Table S1 a”

Comment: Delete space between S1 and a.

A.N.-O.: Done.

Lines 174 “slightly higher compared to the control”

Comment: Here, you should specify what was used as a control (non-transgenic null segregants, according to line 537)

A.N.-O.: Thank you. Added.

Line 175 “Among 65 T2 analyzed plants”

Comment: 2 in T2 should be in lowercase.

A.N.-O.: Thank you. Changed.

Line 177-178 “Additionally 18 plants were not- silenced”

Comment: You need comma after “Additionally” and no space between “not-” and “silenced,’

A.N.-O.: Thank you. Changed.

Figure 1.

Comment: Generally, the top of each chart should be in the middle of the bars. In you diagram the bars look strange as if they are on the top of the column. Would you kindly explain why?

Please, put the letter (a, b, c, …,e) in the high left corner of each diagram as it would be more convenient to find it wheat one is reading the figure.

Please, specify what is shown in x-axis.

It would be a good idea to show the thresholds (0.6 and 0.8) that you used to classify the produced plants in the diagrams.

A.N.-O.: Figure 1 is corrected and description is completed.

Line 188 “and TaCKX2.2.2 vs. TaCKX2.2.1

Comment: according to the diagram, it should be “and CKX2.2.2 vs. CKX2.2.1 and CKX2.1 (e)”

A.N.-O.: Corrected.

Here, the general comment: in some places you use TaCKX, in others, CKX. For example, in Figure 1 there is inconsistencies in gene designation between Figure caption and the diagram names in prefix “Ta-“. In sections 2.1 and 2.2 you describe the genes as CKXs, while in 2.3 and 2.4 as TaCKXs. Please, make it uniform throughout the manuscript where CKXs are described specifically as wheat genes.

A.N.-O.: Thank you. The full name of TaCKX is used/corrected in the text and figure legends and abbreviated names “CKX” are in the figs (to make them clearer).

Lines 196-206

Comment: Here, the TaCKXs genes should be given in italics. And again, please, use TaCKX (not CKX where wheat genes are considered) designation throughout the whole manuscript as different titles hamper understanding the text.

A.N.-O.: Thank you. Corrected.

Figure 3

Comment: What is shown in “CKX act” chart? Please, decipher in Figure caption and add the description of this column in the respective abstract.

A.N.-O.: Thank you. Corrected.

Line 223-224 “trace amounts (≤ 0.02 ng/g biomass…, DZ, …. Not detected: DZ,…”

Comment: Here is inconsistence: was DZ not detected or detected in trace amounts?

A.N.-O.: DZ was not detected. Removed from trace amounts.

Figure 5.

Comment: It is a good idea to use symbols to designate the relative expression level such as = (not changed significantly), =↓ (slightly silenced, not significant), ↓ (silenced, significant), ↓↓ (highly silenced, significant), ↑ (expression increased). So, please, use them uniformly in Figure 3, Figure 5, Table 1, Figure 6. In Figure 3 you may provide the explanation of these symbols in statistical language (significant/ not significant, p<0.05) and hereafter use them in following tables and figures. In particular, replace “nc” by “=” in Figure 5.

A.N.-O.: Thank you. Uniform symbols were included in Fig. 5, Fig. 6, Table 1 and explanations of the symbols are in the legends. The legend in Fig. 3 is corrected, but there was no need to provide explanations of these symbols here.

Lines 230-231

Comment: The text is disrupted here. Please, correct the text makeup

A.N.-O.: Makeup of the text is corrected.

Table 1

Comment: I suggest that you use up and down arrows instead of “-“ in “cc phytohormones” column. It will demonstrate the negative and positive feedback better.

A.N.-O.: This is good suggestion. Corrected.

In TaCKX2.2.2↓↓ you use parentheses for (BA 0.58) and exclamation mark for ABA 0.83! Would you please explain why.

A.N.-O.: Removed.

Figure 6 “-CKX2.1”, “-CKX2.2.1↓”, “-CKX2.1”

Comment: What does “-” (minus) mean here? Since it decreased you do not need minus. It really mislead and should be explained.

A.N.-O.: Thank you. “-“ are removed.

Line 305 Discussion

Comment: Here, at the beginning of Discussion, as well as in the end of Introduction you should briefly mention that you in your previous work you silenced TaCKX1 and this is (to certain extent) a continuation of that research. In this context, the discussion will be more easily comprehended. You here and then reference to your research [11] without mentioning that it is your study. I find it correct to reference to it openly, as it makes the results of [11] and the present study comparable and clear to understand.

A.N.-O.: Yes, I underlined our earlier work with TaCKX1 done with the same genotype in introduction and discussion.

Line 363. 3.3. Homeostatic feedback loop between TaCKX genes transcription and phytohormones

Comment: The idea of feedback loop is great. I suggest that you provide a scheme/model with arrows that demonstrate these feedback loops that proved homeostatic state in plant. For example: TaCKX2.2.2, TaCKX2.2.1, TaCKX1 down -> cZ and iP up-> TaCKX5 up->IAA down. You may develop this scheme for plant based on your and literature results as it may really contribute to the whole understanding of such links between genes-hormones nets.

A.N.-O.: This article is pretty long. Please let me to do this in the next one with results obtained in another cultivar.

Lines 406-441, lines 442-489

Comment: the paragraphs are too large. I suggest that you divide them into several paragraphs according to the main ideas in them.

A.N.-O.: Yes, the first paragraph you mentioned (lines 406-441, now 433-470) is large, because there is discussion on one trait (TGW). However, it was divided to two paragraphs. The second one (lines 442-489, now 471-518) includes two paragraphs.

Line 420 “down regulates”

Comment: the space is excess, “downregulates”

A.N.-O.: Thank you. Corrected.

Lines 421-428: “Our data are not in agreement with those reported for TaCKX6 [36], renamed as TaCKX2.2.1-3D [12]. An 18-bp deletion in intron 2 of the gene resulted in its decreased expression, which was associated with a greater TGW, compared to its wild-type haplotype. However, these and our data are not well comparable because of different nature of these genetic modifications in the genomes. The results of Zhang et al. [36] refer specifically to the mutation of one homoeologous gene, and in 427 our case all three homoeologous genes were downregulated.”

Comment: In sentences before and after this fragment you discuss TaCKX2.2.2 and TaCKX1. But here you discuss TaCKX2.2.1. It confuses a little and makes it hard to follow the discussion. Moreover, you write that “Our data are not in agreement”. But in your experiments TGW slightly (insignificantly) decreased in T1 TaCKX2.2.1-silinced plants and in T2 you TaCKX2.2.1 was silenced along with TaCKX2.2.2 and TGW increased. What particular results are not in agreement? Please, specify.

A.N.-O.: You are right. This is another gene; therefore the discussion was changed starting from line 449: “Opposed data….” and new paragraph was set.

Line 453 “longer grain felling”

Comment: “longer grain filling” should be instead

A.N.-O.: Thank you. Corrected.

Line 546 vs line 554 Ref2/ref2

Comment: Please, use the same designation lowercase or uppercase letter.

A.N.-O.: Thank you. Corrected.

Line 538 4.4 4.4. Quantitative RT-qPCR

Comment: What was the technical replicate?

A.N.-O.: The answer in the text is: (repeated RT-qPCR reaction).

What was the PCR efficiency? Please, demonstrate the calibration curves for the used primers.

Line 536-537 “Non-transgenic, null segregants were used as control samples in all experiments.’

Comment: How many plants were used as a control?

A.N.-O.: These numbers are included in Table s1 and Fig. 2.

Line 565 “tZ”

Comment: Please, decipher (other substances are deciphered here) Thank you. Corrected.

References

Line 671 “Arabidopsis thaliana” should be in italics

Line 730 “Triticum-Aestivum” should be “Triticum aestivum” in italics

A.N.-O.: Thank you. Italics in references are corrected. 

Supplementary Table S2

Comment: I cannot catch the idea of grey highlighting and asterisks (*) in the heads of the table. Would you please specify this?

A.N.-O.: Thank you. Explanation is included “Nonparametric tests are highlighting in grey.” Asterisks are removed.

Reviewer 3 Report

The Authors reported on the influence of the disruption of two TaCKX2.2 genes expression using RNAi technology on some agronomical traits of wheat. The impact of these agronomical traits can be related to a wide reprogramming of phytohormone contents as a consequence of the TaCKX2.2 genes silencing. The Authors also showed that TaCKX2.2 genes can coordinate the expression of some members of the TaCKX gene family.

The paper is well prepared. In particular really appreciate the high quality of the Discussion. I have only very few minor suggestions, mainly concerning the quality of the figures that should be improved.

  • Figure 1 can be improved. For example, add a line corresponding the expression level of control line with relative expression level set at 1. Statistical differences with control expression level should be indicate by *.
  • Figure 2: statistical analysis using different letters in order to show the different statistical groups should be preferred.
  • Figure 4a: add statistical analysis.

Author Response

Dear Reviewer (3),

Thank you very much for your comments and acceptance of the manuscript for publication with minor correction concerning Figure 1, Figure 2 and Figure 4a. We made all corrections as suggested:

Figure 1 – “a line corresponding the expression level of control line with relative expression level set at 1” is added.

Figure 2: “statistical analysis using different letters in order to show the different statistical groups” was applied.

Figure 4a: statistical analysis is added.

Please note that all Figures were corrected and replaced with previous ones.

Best regards,

Anna Nadolska-Orczyk

Reviewer 4 Report

This manuscript showed the effect of RNAi for TaCKX2.2.2 gene on the agronomic trait. My major field is RNA silencing and not for phytohormones and their effects on agronomic trait. So my major queries are forcused on RNA silencing.

(1) Why the degree of RNA silencing on TaCKX2.2.1 are so different between the T1 and T2 generations? The sequences used in the RNAi construct are derived from the TaCKX2.2.2 gene. As authors mentioned, the sequence identities between TaCKX2.2.2 and TaCKX2.2.1 are 93.3~94.6%. In general, such high similarities between the TaCKX2.2.2 and TaCKX2.2.1 are enough to induce simultaneous silencing of both genes. In fact, in Figure 1(c), clear correlation between the silencing degrees of TaCKX2.2.2 and TaCKX2.2.1 was observed. However, in Figure 1(e). such correlation has been lost in the T2 generation. This result may indicate that the TaCKX2.2.1 expression was not correlated with the silencing of TaCKX2.2.2. 

(2) Because the RNAi-inducing sequence is originated from TaCKX2.2.2, the results and discussion regarding to the T1 with silenced TaCKX2.2.1 (Fig.2b) are meaningless, becuase these plants showed lower expressioin of TaCKX2.2.1 and relatively normal expression of TaCKX2.2.2. Therefore these variation in agronomic trait was independent of RNA silencing, perhaps may be caused by regeraration-dependent variation during in vitro culture.

(3) The explanation for figures are not enough for most readers. The errar bar is SD ? SE?  The sample number was absent. Fig. 2c, the statistically significant differeneces should be shown between control vs silenced plants.

(4) Please provide the sequence identities between the sequences used in RNAi construct and each CKX gene family members. Such data is required for the understanding of Figure 3.

(5) The authors should provide the plant material description. The T1 and T2 plants are selected on the hygromycin-containing medium? How about are the copy numbers of the RNAi transgene in each transgenic strain, etc.

Anyway, in the view of RNA silencing, I cannot accept the data originated from the plants in which TaCKX2.2.1 expression was decreased and TaCKX2.2.2 expression was comparable to the control (referred from line 310 in the text 12), because siRNAs for TaCKX2.2.2 are generated in this study.

Author Response

Dear Reviewer (4),

Thank you for your review, which helped to improve the manuscript.

We corrected the manuscript in all sections referring to the p.1 of your review.

The Figure legends were corrected as stated in p. 3 of the review.

Best regards,

Anna Nadolska-Orczyk

(1) Why the degree of RNA silencing on TaCKX2.2.1 are so different between the T1 and T2 generations? The sequences used in the RNAi construct are derived from the TaCKX2.2.2 gene. As authors mentioned, the sequence identities between TaCKX2.2.2 and TaCKX2.2.1 are 93.3~94.6%. In general, such high similarities between the TaCKX2.2.2 and TaCKX2.2.1 are enough to induce simultaneous silencing of both genes. In fact, in Figure 1(c), clear correlation between the silencing degrees of TaCKX2.2.2 and TaCKX2.2.1 was observed. However, in Figure 1(e). such correlation has been lost in the T2 generation. This result may indicate that the TaCKX2.2.1 expression was not correlated with the silencing of TaCKX2.2.2. 

A.N.-O.: The possible reasons for differences in expression level/silencing of TaCKX2.2.1 between T1 and T2 generations are discussed in the manuscript. The main explanation is that the GFM genes are coregulated. We would like to underline, that these data are experimental, and our role is to explain them. Similar coregulation of expression of genes from the same family were reported by us in earlier report (item 11 in references, Jablonski et al. 2020).

As you can see in Table S2a, the correlation between TaCKX2.2.1 and TaCKX2.2.2 in T2 was negative (-0.66), however not significant (opposed to T1).

Concluding we can only say that we did the experiments, reported the results, and did the statistical analysis, which showed the correlation patterns.

(2) Because the RNAi-inducing sequence is originated from TaCKX2.2.2, the results and discussion regarding to the T1 with silenced TaCKX2.2.1 (Fig.2b) are meaningless, becuase these plants showed lower expressioin of TaCKX2.2.1 and relatively normal expression of TaCKX2.2.2. Therefore these variation in agronomic trait was independent of RNA silencing, perhaps may be caused by regeraration-dependent variation during in vitro culture.

A.N.-O.: Thank you for this suggestion, however we do not thing that these differences are the effect of in vitro culture. All silenced plants showed similar pattern of changes and we included regenerated plants as the control.

(3) The explanation for figures are not enough for most readers. The errar bar is SD ? SE?  The sample number was absent. Fig. 2c, the statistically significant differeneces should be shown between control vs silenced plants.

A.N.-O.: Thank you for this comment. SD and SE were added in Figure legends. The number of samples are also included (in brackets of the Figure 2). The statistically significant differences are marked additionally with the letters.

(4) Please provide the sequence identities between the sequences used in RNAi construct and each CKX gene family members. Such data is required for the understanding of Figure 3.

A.N.-O.: Thank you for the suggestion. We added Table S4 to supplementary materials entitled: “Sequence queries (%) and identities (%) between the sequence used in RNAi construct and each TaCKX gene family member located to the genomes A, B and D”.

(5) The authors should provide the plant material description. The T1 and T2 plants are selected on the hygromycin-containing medium? How about are the copy numbers of the RNAi transgene in each transgenic strain, etc.

A.N.-O.: T0 plants, in vitro regenerated, were selected on hygromycin-containing medium.

“Putative transgenic events as well as T1 plants were verified by PCR screening for the presence of a T-DNA fragment using primers specific to the hpt gene.” This set of information is included in the Materials and Methods section.

Hygromycin selection of T1, T2 plants (as implied from the questions) can be done only working with Arabidopsis. The seedlings are small enough to grow in the medium supplemented with hygromycin. This approach however is completely unfeasible with wheat/barley seeds and wheat/barley seedlings, because of the seeds and seedlings size and growth in soil not MS medium.

Anyway, in the view of RNA silencing, I cannot accept the data originated from the plants in which TaCKX2.2.1 expression was decreased and TaCKX2.2.2 expression was comparable to the control (referred from line 310 in the text 12), because siRNAs for TaCKX2.2.2 are generated in this study.

A.N.-O.: In response to this remark, we would like to underline that we present experimental data. It is beyond the scope of this letter to discuss the differences between the RNAi-based silencing in Arabidopsis and in wheat (cereals). Our long experience with transgene expression and experimental gene silencing in wheat and barley indicates either transgene expression or gene silencing may change in the subsequent generations. It is also not uniform in plants from each progeny population. This is clearly different what can be found in Arabidopsis. One of the possible explanations is the much bigger size (and complexity) of wheat genome (ab. 17 Gb; more than 100 times bigger) comparing to Arabidopsis (135 Mb).

Round 2

Reviewer 1 Report

Addressed my concerns.

Author Response

Dear Reviewer (1),

Thank you for accepting our revisions and explanations.

Best regards,

Anna Nadolska-Orczyk

Reviewer 4 Report

The revised manuscript was much improved. However, in the view of RNA silencing, the results cannot be explained by the simple RNA silencing effects. I strongly recommend to revise (minor revision) as follows.

(query) Why the degree of RNA silencing on TaCKX2.2.1 are so different between the T1 and T2 generations? The sequences used in the RNAi construct are derived from the TaCKX2.2.2 gene. As authors mentioned, the sequence identities between TaCKX2.2.2 and TaCKX2.2.1 are 93.3~94.6%. In general, such high similarities between the TaCKX2.2.2 and TaCKX2.2.1 are enough to induce simultaneous silencing of both genes. In fact, in Figure 1(c), clear correlation between the silencing degrees of TaCKX2.2.2 and TaCKX2.2.1 was observed. However, in Figure 1(e). such correlation has been lost in the T2 generation. This result may indicate that the TaCKX2.2.1 expression was not correlated with the silencing of TaCKX2.2.2.

(author reply): The possible reasons for differences in expression level/silencing of TaCKX2.2.1 between T1 and T2 generations are discussed in the manuscript. The main explanation is that the GFM genes are coregulated. We would like to underline, that these data are experimental, and our role is to explain them. Similar coregulation of expression of genes from the same family were reported by us in earlier report (item 11 in references, Jablonski et al. 2020).

As you can see in Table S2a, the correlation between TaCKX2.2.1 and TaCKX2.2.2 in T2 was negative (-0.66), however not significant (opposed to T1).

Concluding we can only say that we did the experiments, reported the results, and did the statistical analysis, which showed the correlation patterns.

(reviewer’s recommendation) As authors mentioned, the expression level of TaCKX2.2.2 and TaCKX2.2.1 cannot be explained by the “orthodox” silencing mechanisms which have been determined using Arabidopsis and other model plants. If so, the authors have to mention this point more clearly in section 3.1, becuase most readers might confuse as follows:  “Why the siRNAs targeting the TaCKX2.2.2 cause the weak silencing of TaCKX2.2.2 and strong silencing of TaCKX2.2.1?” 

Author Response

Dear Reviewer (4),

Thank you for your recommendation. As we mentioned before, mechanisms of silencing in model plants differ significantly from that, which occur in polyploids (in our case hexaploid wheat).

This is because gene regulation is much more complex in polyploids, as described for example in: Cheng, F.; Wu, J.; Cai, X.; Liang, J. L.; Freeling, M.; Wang, X. W., Gene retention, fractionation and subgenome differences in polyploid plants. Nat Plants 2018, 4, (5), 258-268 and also many others.

Please accept that we will not discuss these differences (ie. between model Arabidopsis and polyploid cereal) in the discussion section, because it would be out of scope of the manuscript which is focused entirely on wheat. This would be also somehow strange for researchers working with polyploids, who are potentially the main group of readers of our paper.

Our data are discussed with other papers on wheat silencing or cereal silencing. We would like to underline that differences in levels of silencing between target genes as well as different silencing of the genes in the subsequent generations of tested plants are not unusual. They have been reported before also in our earlier articles and they have not been questioned by readers of our papers as well as other three Reviewers of this manuscript.

Best regards,

Anna Nadolska-Orczyk